# Tetrahydroisoquinoline N-methyltransferase from *Methylotenera* Is an Essential Enzyme for the Biodegradation of Berberine in Soil Water

**DOI:** 10.3390/molecules27175442

**Published:** 2022-08-25

**Authors:** Runying He, Yao Cui, Ying Li, Xizhen Ge

**Affiliations:** College of Biochemical Engineering, Beijing Union University, Beijing 100023, China

**Keywords:** berberine, soil water, *Methylotenera*, tetrahydroisoquinoline N-methyltransferase, biodegradability

## Abstract

Berberine (BBR), a Chinese herbal medicine used in intestinal infection, has been applied as a botanical pesticide in the prevention of fungal disease in recent years. However, its degradation in the environment remains poorly understood. Here, we investigated BBR’s degradation in soil water from different sources accompanied by its effect on bacterial diversity. Our results indicated that BBR was only degraded in soil water, while it was stable in tap water, river water and aquaculture water. Bacterial amplicon results of these samples suggested that the degradation of BBR was closely related to the enrichment of *Methylotenera*. To reveal this special relationship, we used bioinformatics tools to make alignments between the whole genome of *Methylotenera* and the pathway of BBR’s degradation. An ortholog of Tetrahydroisoquinoline N-methyltransferase from plant was discovered only in *Methylotenera* that catalyzed a crucial step in BBR’s degradation pathway. In summary, our work indicated that *Methylotenera* was an essential bacterial genus in the degradation of BBR in the environment because of its Tetrahydroisoquinoline N-methyltransferase. This study provided new insights into BBR’s degradation in the environment, laying foundations for its application as a botanical pesticide.

## 1. Introduction

Berberine (BBR, Figure 1) is a plant-derived isoquinoline alkaloid widely distributed in plants of the Berberidaceae, Ranunculaceae and Papaveraceae families [1]. BBR has been used in anti-tumor disease, anti-oxidative stress, anti-Alzheimer’s disease, myocardial tissue protection and anti-hyperglycemic drugs [2,3,4]. Today, its application is extended as a botanical pesticide because of its exceptional antifungal effect [5,6]. The MIC of BBR against *Monilinia fructicola* is as low as 4.5 µg/mL according to our previous work, showing the potential to prevent fungal disease in peach [7]. With rapidly elevated demand of BBR in agricultural applications, increasing BBR residues remain in the soil and even filter into the underground water [8]. Therefore, the impacts of BBR on the environment need to be further elucidated.

The degradation of BBR in soil was studied in our previous work [6]. We demonstrated that BBR can be completely degraded in soil within 9 days, and soil bacterial diversity was naturally restored as BBR degraded [6]. However, the retention time of BBR in soil was not as long as 9 days in nature, and some residues were filtered into the soil water or even into underground water [9]. When BBR enters water through runoff, leaching and wastewater discharge [8,10], the degradation of BBR in water and its impact on bacterial diversity in the water environment need further investigation. It has been confirmed that the microorganisms that can degrade BBR include *Sphingopyxis* [11], *Burkholderia* [12], *Rhodococcus* [13] and *Bacillus* [6]. Mixed cultures of *Hydrogenophaga*, *Azoarcus*, *Sphingopyxis*, *Stenotrophomonas*, *Shinella*, *Alcaligenes* and *Nitrospir* are identified as potential functional species for the biodegradation of BBR and its metabolites [14]. However, the degradation pathway of BBR in bacteria is still unclear. According to the secondary metabolite synthesis pathway in plants (Kyoto Encyclopedia of Genes and Genomes (KEGG): map00950, map01063), BBR’s degradation may rely on multiple steps, while these enzymes are deduced as not being contained by a single bacterium. Therefore, the digestion of BBR in microorganisms needs further study.

The diversity of microbial populations plays an important role in environmental remediation and pesticide degradation [15,16]. The improvement of the microbial community can significantly increase the degradation efficiency of pesticides in the environment [17]. In the case of BBR, there is a possibility that the enzymes essential for the degradation of BBR are distributed in different microorganisms, requiring them to participate in BBR’s degradation simultaneously. On the other hand, there is also a possibility that there is a rate-limiting enzyme that exists in only one or a few microorganisms. When key enzymes are present, the degradation process will be accelerated significantly, and this phenomenon can be also found in the degradation of lignin [18].

Based on these backgrounds, in this study, we aim to investigate BBR’s degradation in soil water from different sources and explore its relationship with bacterial diversity. We anticipate revealing the relationship between BBR’s degradation and the changes in the bacterial content in soil water, revealing the essential step of BBR’s degradation in the environment.

## 2. Materials and Methods

### 2.1. Materials

BBR (99%), methanol and phosphoric acid were purchased from Macklin (Shanghai, China). Dimethyl sulfoxide (DMSO) was purchased from Bailens Biotechnology Co., Ltd. (Tianjing, China). DNA primers (Miseq-F and Miseq-R) used for bacterial identification were synthesized by Shanghai Hanyu Biotechnology Co., Ltd. The genomes of each sample were extracted, and then amplicon sequencing was conducted using the two primers Miseq-F:TCGTCGGCAGCGTCAGATGTGTATAAGAGACAGTACGGRAGGCAGCAG and Miseq-R: GTCTCGTGGGCTCGGAGATGTGTATAAGAGACAGAGGGTAT CTAATCCT. DNA sequencing was conducted in an Illumina Miseq platform. DNA polymerase and the other agents used for 16s rDNA amplification were products of Takara (Dalian, China).

### 2.2. Collection and Treatment of Water in Different Environments

Soil samples with a depth of 5 to 10 cm from different locations of China (Xinjiang, Beijing, Jiangxi, Yunnan, Heilongjiang and Guangxi) were collected and mixed evenly. Then, 50 g of the soil was soaked in 500 mL of sterilized water and filtered through a 22 µm filter to eliminate the insoluble particles. Subsequently, 500 mL of fish-cultured water, river water and tap water were respectively filtered and used in the following experiments. BBR was dissolved in DMSO as 10 mg/mL as a stock solution, and the final concentration of BBR in the experiment was set at 100 µg/mL.

### 2.3. HPLC Analysis of BBR

The standard BBR sample used in HPLC analysis was dissolved in sterile water to the final concentration of 100 µg/mL, and gradient concentrations of BBR samples were diluted and filtered through a 0.22 µm filter membrane before they were injected into the HPLC system. For each sample obtained from the soil water, samples were filtered before injection. The HPLC system (Shimazu, Kyoto, Japan) was equipped with a C_18_ column with a SPD-20A UV detector at 364 nm wavelength. The column temperature was 30 °C, and the injection volume was 20 μL. The mobile phase was composed of methanol (50%) and 0.05% phosphoric acid water (50%), and the flow rate was set at 0.8 mL/min. This is a commonly used method in our lab [6]. Data in the results were presented as means ± standard errors of mean (SD). Statistical significance was indicated by P values analyzed by one-way ANOVA.

### 2.4. Degradation Experiment of BBR in Water Samples

Each 20 mL sample of soil water, fish-cultured water, river water and tap water was collected, and the initial concentration of BBR was set at 100 µg/mL. All samples were placed in a constant temperature incubator at 30 °C for 14 days. Samples were taken every 24 h and centrifuged at 12,000 rpm for 5 min, and then filtered through a 0.22 µm membrane for HPLC detection.

### 2.5. 16s rDNA Sequencing of Soil Water Samples

The genomes of samples from Xinjiang, Beijing, Jiangxi, Yunnan, Heilongjiang and Guangxi were collected and then extracted for amplicon sequencing. Briefly, 1 mL of the soil water samples was obtained from the culture of BBR degradation. The total genome of the soil water samples was extracted using a whole genome extraction kit from Tiangen, China. Next, 2 μL of genome was used as the template for amplicon sequencing. The other agents were added as follows: 25 μL of Ex Taq, 6 μL of Miseq F, 6 μL of Miseq R, 11 μL of sterile water. PCR was carried out as follows: 94 °C for 10 min, 30 cycles (94 °C for 1 min, 55 °C for 40 s, 72 °C for 2 min), 72 °C for 1 min, and 16 °C hold. PCR amplification products were detected by 1% agarose gel electrophoresis, and then were sent to Shanghai Hanyu Biotechnology Co., Ltd. for 16s rDNA amplicon sequencing. The content of each genus was calculated by comparing the number of sequences identified belonging to certain genera and the total number of sequences in the same sample.

Soil water samples (Xinjiang, Beijing and Jiangxi) at 0 days, 3 days, 6 days, 10 days and 14 days were cultured in Luria-Bertani (LB) agar plates (NaCl 10 g/L; peptone 10 g/L; yeast extract 5 g/L; agar powder 15 g/L, if possible) at 30 °C for 24 h. Possible BBR-degrading strains with different colonial morphologies were collected and characterized. After purification three times, the genome of the strain was extracted, and 16s rDNA was amplified and sequenced by Sanger sequencing. All the strains were conserved in −80 °C mixed with 15% sterile glycerol.

### 2.6. Degradation of BBR by Isolated Bacterium

Soil waters were sterilized and then cooled for later use. Strains were grown overnight in LB with a 30 °C shaking incubator (200 rpm) prior to experiment. For the degradation experiment with single and multiple bacterial: 10 mL of the recovered single or mixed bacteria was washed three times and then added to 100 μL of sterilized water, followed by adding 50 μL of each to soil water or LB medium.

Next, the degradation experiment was conducted in the shake flask in the presence of 100 µg/mL BBR. The samples were shaken (200 rpm, 37 °C), and samples were taken at 0 day, 4 days, 8 days, 10 days and 14 days for HPLC analysis.

### 2.7. Genome Alignment and Homology Modeling

The biosynthesis pathway of BBR in plant (KEGG) was used as the template for detecting enzymes that participated in the degradation of BBR bacteria. Briefly, the enzymes in plants were collected, and the bacteria genomes were prepared for alignment. Then, the enzymes from BBR biosynthesis in plants were used as the template, and the alignment was conducted with the whole genome of each isolated bacterium. The genome alignment was carried out in Python, and the script used in this study was available from Github (https://github.com/liying0128/pathway_finder (accessed on 12 July 2022). Homology modeling was generated from YASARA Structure since we have the license. According to the protein sequences from plant and bacteria, models were generated and visualized after alignment and energy minimization.

## 3. Results

### 3.1. BBR Is Degraded Only in Soil Water

In order to evaluate the effect of BBR on the environment as agricultural applications, the degradation of BBR was carried out on samples of soil water, fish-cultured water, river water and tap water (Figure 2A). The results indicated that stabilities of BBR were distinct in different samples, and BBR’s degradation was only observed in samples of soil water. In soil water, the adaptation period of BBR was nearly 5 days before degradation, while BBR degrades rapidly, within 3 days after the initiation of degradation. It is speculated that the content of microorganisms in the soil water may be rich and that there are single or multiple bacteria that can favor the degradation of BBR.

To clarify whether the degradation of BBR by soil water is universal, BBR degradation experiments were carried out in soil water from different regions of China (Xinjiang, Beijing, Jiangxi, Yunnan, Heilongjiang and Guangxi; Figure 2B). Surprisingly, only in three soil water samples was BBR degraded. Notably, the adaption times of BBR degradation in these samples were different, while BBR degraded rapidly within two days after the degradation began. Therefore, it is concluded that the degradation of BBR in soil water is not universal, and it is consistent with Strickland’s conclusion that soil microbial communities in different regions are functionally different [19].

### 3.2. The Effect of BBR on Bacterial Diversity of Soil Water

Since the degradation of BBR in soil water was different among these samples, we conducted amplicon sequencing of the bacterial diversity on different days to investigate the variations in the contents of the bacteria during BBR’s degradation.

The most significant differences between the samples were the contents of *Methylotenera* and *Novosphingobium* in soil water. At day 0, the bacterial content of both *Methylotenera* and *Novosphingobium* was nearly 0%. Interestingly, in the soil water in which BBR was degraded, the two strains were both enriched significantly within 6 days. When *Methylotenera* reached the peak on day 3, the content of *Novosphingobium* began to increase (Figure 3A). However, in the sample in which BBR was not degraded, the content of these two strains did not change in the first 6 days. Compared with the different degradation rates of BBR, we suspect that the enrichments of *Methylotenera* and *Novosphingobium* had positive effects on the degradation of BBR.

In contrast, *Flavobacterium* genus in both groups began to grow on day 0 and reached a peak on day 3, but its content in the samples from Xinjiang, Beijing and Jiangxi was two times higher than that in the samples from Yunnan, Heilongjiang and Guangxi. However, its content decreased from day 3 to the end (Figure 3B). It is deduced that the presence of BBR stimulated the enrichment of *Flavobacterium*. In parallel, the bacterial contents of *Delftia* and *sphingopyxis* were nearly 0% in the initial of the experiments. In the samples from Xinjiang, Beijing and Jiangxi, the contents of these two strains began to increase after 3 daysof cultivation, and the increasing rate was even faster after 6 days, while its content in the samples from Yunnan, Heilongjiang and Guangxi did not change (Figure 3C). Therefore, these two strains might also have positive effects on the degradation of BBR.

The three genera of *Pseudomonas*, *Stenotrophomonas* and *Acinetobacter* were all detected in the initial of the two experiments. The bacterial contents of *Pseudomonas* and *Stenotrophomonas* were decreased in all of the samples. Different from these two strains, the content of *Acinetobacter* decreased rapidly to 0% in the samples from Xinjiang, Beijing and Jiangxi within 3 days, but its contents in the samples from Yunnan, Heilongjiang and Guangxi did not change (Figure 3D).

To sum up, *Acinetobacter* was abundant in initial soil water samples, while in content, it decreased sharply within 3 days in the samples from Xinjiang, Beijing and Jiangxi. In contrast, *Novosphingobium*, *Delftia* and *sphingopyxis* started to grow after day 3. Notably, the bacterial contents of *Methylotenera* increased sharply in the presence of BBR but decreased with the degradation of BBR.

### 3.3. Isolated Bacteria Were Unable to Degrade BBR

To explore whether single or multiple bacteria can degrade BBR in an artificial medium, bacteria were isolated from the soil water in which BBR was degraded, and these bacteria were named J1–8. DNA sequence alignment showed that these strains belonged to three different genera (Table 1)

Next, single- and multiple-degradation experiments (J1–8) were carried out for 14 days in LB liquid medium and sterilized soil water (both contain 100 µL/mL BBR), respectively. The results showed that in LB liquid medium (Figure 4A) and soil-sterilized water (Figure 4B), neither of the single strains could degrade BBR. Next, multiple-degradation experiments (containing all of J1–8) were also carried out in LB liquid medium and soil-sterilized water with 100 μg/mL of BBR. Similar results were obtained and BBR was not degraded in presence of all these strains (Figure 4C,D). Together with the data from amplicon sequencing, we can conclude that the essential bacteria for BBR’s degradation might be unable to be cultivated in artificial medium.

### 3.4. Whole-Genome Alignment Identifies an Essential N-methyltransferase from Methylotenera

Although BBR was degraded in several soil water samples, the initiation ofthe degradation of BBR was different. However, the variations in the bacterial contents of *Methylotenera*, *Novosphingobium*, *Delftia*, *sphingopyxis* and *Flavobacterium* were similar in these soil water samples. Therefore, synthetic pathways of BBR and its related enzymes were obtained from KEGG (Figure 5), and the protein sequences of related enzymes were aligned with the genomes of *Methylotenera*, *Novosphingobium*, *Delftia*, *sphingopyxis* and *Flavobacterium*, the contents of which were increased in BBR’s degradation in soil water. The results indicated that orthologs of enzymes 1, 2, 3, 9 and 24 can be found in these bacteria (protein sequence similarity >30%).

At the same time, these protein sequences were also aligned to the isolated bacteria from soil water (J1−8; these bacteria could not degrade BBR). The results showed that these bacteria contained orthologs or isoenzymes of enzyme 1, 2, 3 and 24.

Surprisingly, the ortholog of enzyme 9 can only be found in *Methylotenera.* Therefore, we can deduce that this enzyme is an essential enzyme in the degradation of BBR, which catalyzes three different reactions and existed solely in *Methylotenera* [20].

The sequence alignment of Tetrahydroisoquinoline N-methyltransferase (Ortholog of enzyme 9) from *Methylotenera* indicated a conserved region of GCGXG (Figure 6) [20]. The results are consistent with the study by Torres et al., who found that GXGXG is a highly conserved region [20]. More importantly, this conserved region is located at the active site (Figure 6) [20,21]. In addition, the protein structures and active site of the enzymes from plant and *Methylotenera* were similar based on the structures generated in silico (Total RMSD value of 5.03, Figure 7). Therefore, Tetrahydroisoquinoline N-methyltransferase from *Methylotenera* might have an essential role in the degradation of BBR.

To explore the distribution of Tetrahydroisoquinoline N-methyltransferase in bacteria at the genus level, the protein sequence of the enzyme was blasted and visualized (Figure 8). The top 500 bacteria were selected for analysis according to the sequence similarity, and there were 219 bacteria with similar sequences among the bacteria at the genus level. Among them, there were 11 strains of *Methylotenera*, accounting for 5.02% of the total. In addition, the largest proportion of orthologs was also identified in *Variovorax*, which have been reported to regulate the homeostasis of plant hormones and restore contaminated soil and water [22,23].

## 4. Discussion

In this work, the degradations of BBR in water from different sources and in soil water from different regions are investigated. The 16s rDNA amplicon sequencing is carried out, and the relationship between BBR’s degradation and bacterial diversity is explored. In addition, the protein sequences of enzymes in the synthesis pathway of BBR are aligned with the whole genome of related bacteria. We demonstrated that Tetrahydroisoquinoline N-methyltransferase, which catalyzes three reactions in the BBR synthesis pathway, was discovered in *Methylotenera,* the content of which had positive relationships with BBR’s degradation. As a highly conserved protein, Tetrahydroisoquinoline N-methyltransferase is deduced to be an essential enzyme in the degradation of BBR by bacteria in soil water.

The degradations of BBR were distinct in soil water collected from different places, while the bacterial contents of *Methylotenera*, *Novosphingobium*, *Delftia*, *Sphingopyxis* and *Flavobacterium* were all increased in the samples in which BBR was degraded. Due to the long synthetic pathway of BBR in plants, one bacterium is unlikely to contain all the orthologs that can be used for the degradation of BBR. Therefore, the bacterial degradation of BBR is mainly accomplished by the bacterial community. However, BBR is degraded in only three soil water samples, and it is stable in the other ones within the 14-day cultivation. More importantly, once BBR is initiatedfor degradation, it will be consumed rapidly within two days. These results provide strong evidence that there exist one or more degradation-limiting steps in bacteria community. Even though several bacteria are isolated and grown in the artificial medium, BBR’s degradation is not observed when cultivated with these strains. *Methylotenera* might be a genus that is unable to grow in the artificial medium, but our data suggested that it is an important element in the degradation of BBR in soil water because of its Tetrahydroisoquinoline N-methyltransferase.

Another aim of our work is to answer whether the application of BBR is beneficial to the soil bacterial diversity or not. Among the strains with increased bacterial contents under the treatment of BBR, most of them are beneficial to the environment except *Flavobacterium*. Among them, *Methylotenera* exists belongs to *Methylophilaceae*. *Methylophilaceae* consists of two species, *Methylotenera mobilis* and *Methylotenera versatilis*, both of which were isolated from Lake Washington sediments [24]. The study reported that *Methylotenera mobilis* is an obligate methylamine utilizer. *Methylotenera versatilis* does not possess the hallmark methanol dehydrogenase of *Methylophilaceae*, but it still showed weak growth on solid agar medium supplemented with methanol [24,25]. It has been confirmed that *Methylotenera* can be used as a degradation agent for environmental pollutants such as petroleum, motor oil and methane [26,27,28]. In addition, *Novosphingobium* is an excellent phenanthrene degrader [29]. *Delftia* can biodegrade organic pollutants, such as phenolic compounds and chlorobenzene [30]. *Sphingopyxis* can efficiently degrade the aromatic compounds, such as microcystins, tetralin, styrene and triclosan [31]. These data demonstrate that BBR’s degradation enriched beneficial bacteria in the environment.

N-methyltransferases participate in the biosynthesis of the monoterpene indole alkaloids vindoline and ajmaline. Based on the widespread presence of a large number of N-methylated and N,N-dimethylated compounds, N-methyltransferases play an important role in Benzylisoquinoline alkaloids metabolism [32]. In particular, Tetrahydroprotoberberine N-methyltransferase exhibits specificity for certain intermediates in the tetrahydroprotoberberine branch pathway. Tetrahydroprotoberberine N-methyltransferase is the only AdoMet-dependent N-methyltransferase in plant alkaloid metabolism [33]. Here we identify an ortholog from bacteria and suggest its importance in biodegradation of BBR. As an enzyme that catalyzes this rate-limiting process, Tetrahydroprotoberberine N-methyltransferase might be a promising enzyme in soil bioremediation.

In summary, we demonstrated the essential factors in BBR’s degradation in soil water, providing evidence of its effects on the environment. Future work will be concentrated on analyzing this important enzyme to provide it as a soil bioremediation agent.

## 5. Conclusions

Extended applications of BBR pose potential risks to the environment. Herein, we investigated BBR’s degradation in water. We confirmed that BBR was only degraded in soil water, while it was stable in tap water, river water and aquaculture water. The degradation of BBR in soil water was closely related to the enrichment of *Methylotenera*. Interestingly, an ortholog of Tetrahydroisoquinoline N-methyltransferase from plant was discovered only in *Methylotenera* and catalyzed a crucial step in BBR’s degradation pathway.

## Figures and Tables

**Figure 1 molecules-27-05442-f001:**
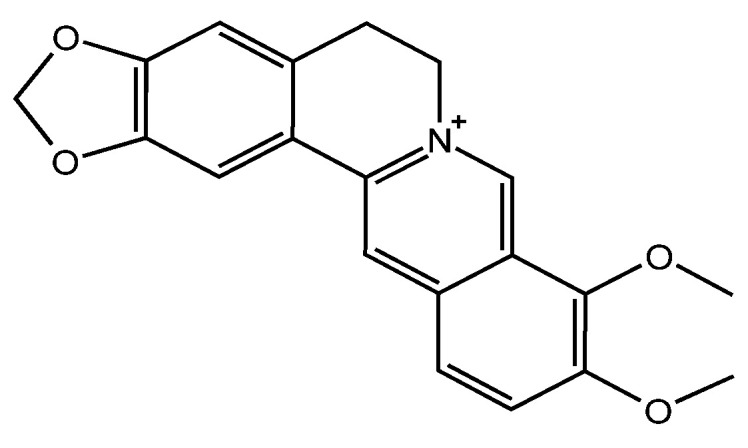
Confirmed Chemical structure of BBR.

**Figure 2 molecules-27-05442-f002:**
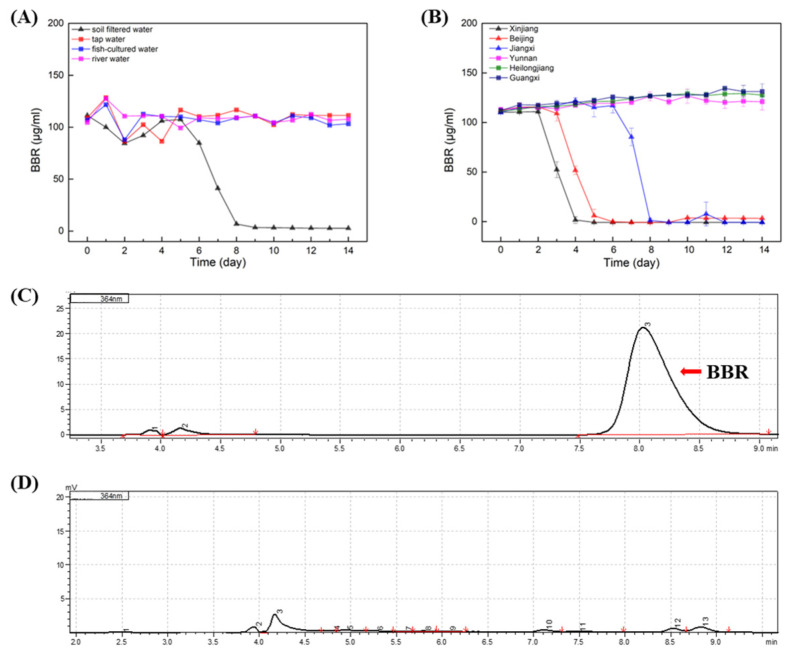
BBR: Berberine; HPLC: High Performance Liquid Chromatography. BBR degradation in different waters. (**A**): Degradation of BBR in soil water, fish-cultured water, river water and tap water. (**B**): Degradation of BBR in soil water samples from different location of China. (**C**): HPLC analysis of BBR’s concentration in the initial of experiment. (**D**): HPLC analysis of BBR’s concentration in the end of experiment. Arrow indicated the peak of BBR.

**Figure 3 molecules-27-05442-f003:**
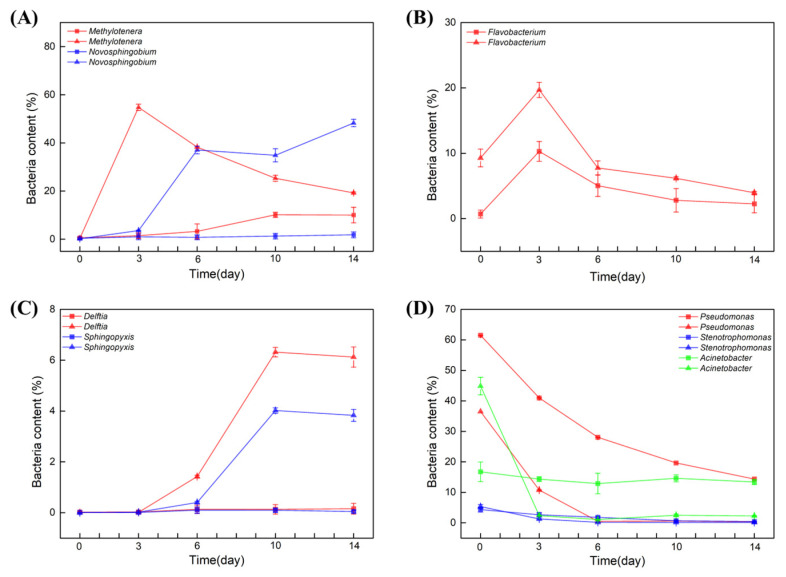
The effect of BBR on bacterial diversity of soil water. Bacterial content within 14 days: *Methylotenera* and *Novosphingobium* (**A**), *Flavobacterium* (**B**), *Delftia* and *sphingopyxis* (**C**), *Pseudomonas*, *Stenotrophomonas* and *Acinetobacter* (**D**). Triangular pattern represents BBR degraded samples. Square pattern represents BBR undegraded samples.

**Figure 4 molecules-27-05442-f004:**
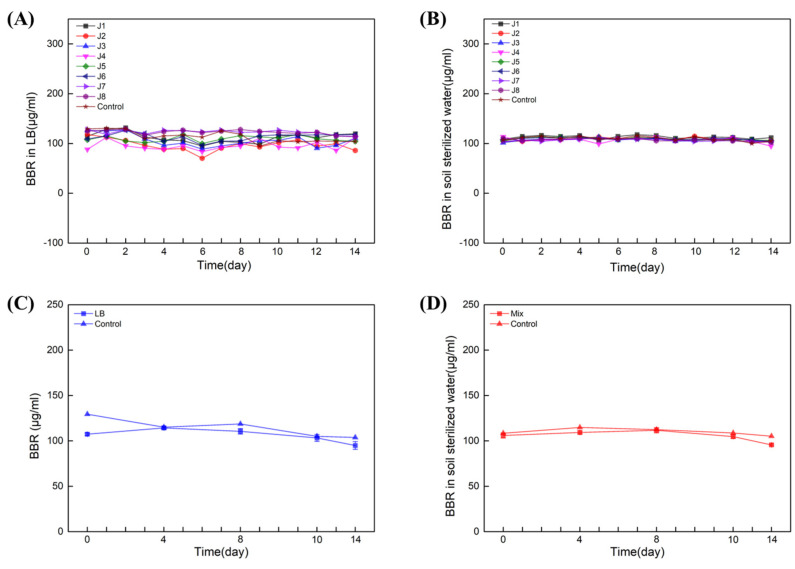
Eight strains failed to degrade BBR. Degradation of BBR by J1−8 in LB: Experiments with single (**A**) and multiple (**C**) degradations. Degradation of BBR by J1−8 in soil sterilized water: Experiments with single (**B**) and multiple (**D**) degradations. Biological replicates were conducted for three times.

**Figure 5 molecules-27-05442-f005:**
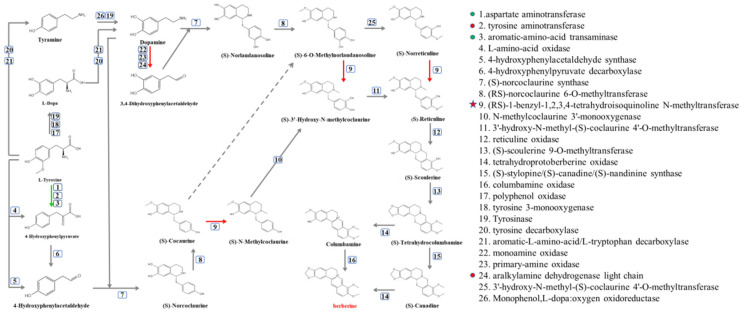
Synthetic pathway of BBR. Green arrows represent the presence of the enzyme in bacteria with increased bacterial content and bacteria isolated from soil water. The red arrow indicates that the enzyme is only present in bacteria with increased bacterial content.

**Figure 6 molecules-27-05442-f006:**
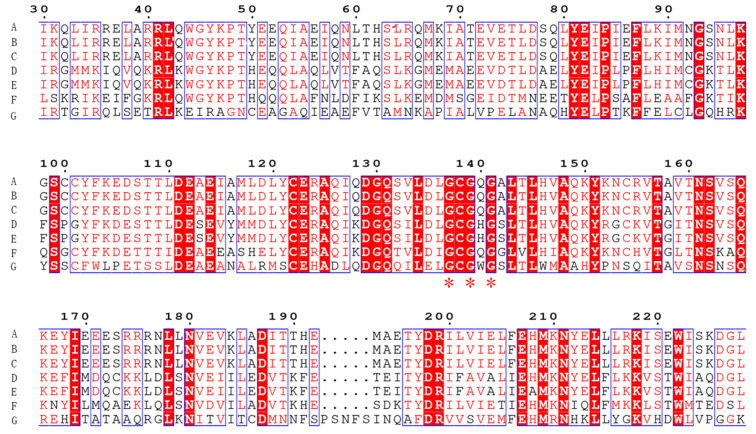
Conserved sequences identified by multiple gene sequence alignments. Alignment of Tetrahydroisoquinoline N-methyltransferase (from the Coptis japonica and from *Methylotenera*) with its similar multiple sequences. Sequences highlighted in red indicate conserved regions. The abbreviations are as follows: A: (RS)-1-benzyl-1,2,3,4-tetrahydroisoquinoline N-methyltransferase, ag:BAB71802 K13384; B: Crystal structure of Coclaurine N-Methyltransferase, PDB: 6gkv; C: Crystal structure of Coclaurine N-Methyltransferase, PDB: 6gkz; D: Pavine N-methyltransferase in complex with Tetrahydropapaverine and S-adenosylhomocysteine, PDB: 5kok; E: Pavine N-methyltransferase H206A mutant in complex with S-adenosylmethionine, PDB: 5kpc; F: Tetrahydroprotoberberine N-methyltransferase in complex with S-adenosylmethionine, PDB: 6p3n; G: cyclopropane-fatty-acyl-phospholipid synthase family protein, WP_013148163.1.

**Figure 7 molecules-27-05442-f007:**
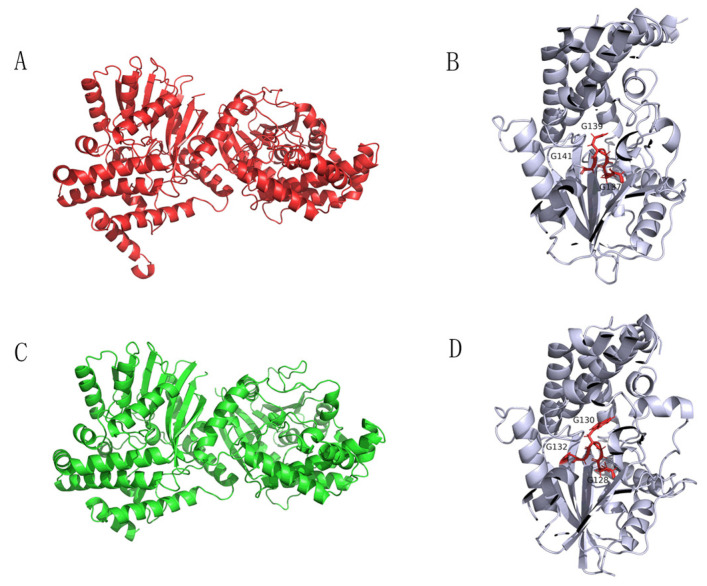
Protein structure diagrams. The proteins were visualized in ribbon representation. Dimeric structure (**A**) and its active site (**B**) of Tetrahydroisoquinoline N-methyltransferase from Coptis japonica. Dimeric structure of Tetrahydroisoquinoline N-methyltransferase in *Methylotenera* (**C**) and its active site (**D**).

**Figure 8 molecules-27-05442-f008:**
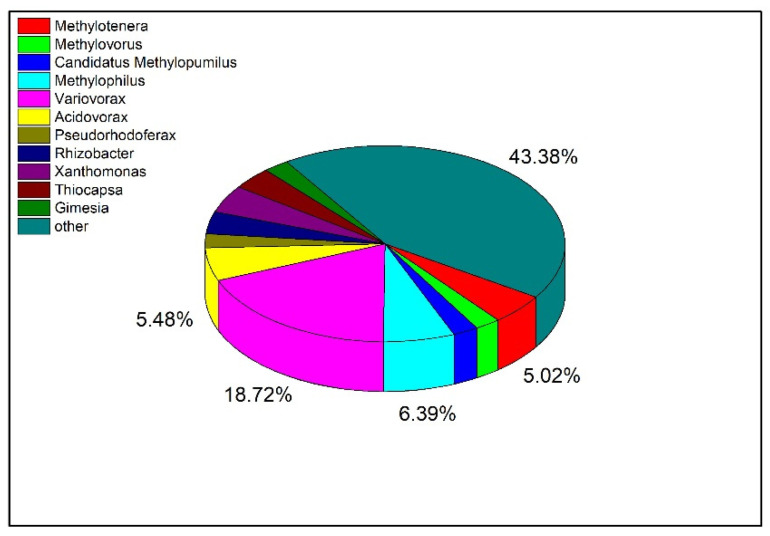
Distribution of Tetrahydroisoquinoline N-methyltransferase in Different Bacteria.

**Table 1 molecules-27-05442-t001:** Bacteria isolated from BBR-degrading soil water.

Bacteria	Genus
J1	*Pseudomonas* sp. confirmed
J2	*Sphingobacterium* sp.
J3	*Pseudomonas* sp.
J4	*Sphingobacterium* sp.
J5	*Agrobacterium* sp.
J6	*Pseudomonas* sp.
J7	*Pseudomonas* sp.
J8	*Pseudomonas* sp.

## Data Availability

The data presented in this study are available on request from the corresponding author.

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
