# Peer review of "Tetrahydroisoquinoline N-methyltransferase from Methylotenera Is an Essential Enzyme for the Biodegradation of Berberine in Soil Water"

_molecules, 2022, doi:10.3390/molecules27175442_

Round 1

Reviewer 1 Report

In the first sentence of the Introduction section, the author placed (Figure 1) after plant families, whereas the figure shows the structure of berberine.

What is antioxidant disease? Is it oxidative stress? The author should write it exactly.

What is the unit for the MIC of berberine against M. fructicola? The author wrote 4.5 only.

In the Introduction section, the author did not show

1) why is the degradation of berberine by bacteria important?

2) why is the presence of berberine in the environment a problem?

On the other hand, if the author suggests degradation of berberine with microorganisms, is it safe, particularly if he describes using pathogenic strains e.g. Shigella sp. or Rhodococcus sp. (a few species are pathogenic).

In the Materials and methods section, the author shows that berberine was analyzed by HPLC  method. Was the chromatographic method validated or was it implemented from another work? Please specify the appropriate reference.

Please, place chromatograms for that analysis in the results section.

Section 2.5. (here, the author wrote incorrectly 2.516 instead 2.5) from what samples the genome was extracted; it is not clear.

Section 2.6. - How bacteria were identified and characterized. Did the author use mass spectrometry? Please show the results for that analysis in the Results section.

The author did not calculate any statistical analysis. Thus his inference is burdened with a significant error.

The preparation of the manuscript is negligent. The methodology does not allow the repetition of the experiments.

Section 3.5 is instead a review than a discussion. In this section, the author should present enzymatic or proteomic analysis results.

Author Response

Reviewer #1:

1.In the first sentence of the Introduction section, the author placed (Figure 1) after plant families, whereas the figure shows the structure of berberine.

Corrected!

2.What is antioxidant disease? Is it oxidative stress? The author should write it exactly.

Corrected!

3.What is the unit for the MIC of berberine against M. fructicola? The author wrote 4.5 only.

Corrected! 4.5 µg/ml

4.In the Introduction section, the author did not show

1) why is the degradation of berberine by bacteria important?

Most of the reported berberine-degrading microorganisms are bacteria. Such as Sphingopyxis sp., Burkholderia sp., Rhodococcus sp. and Bacillus sp.

2) why is the presence of berberine in the environment a problem?

The widespread use of berberine (BBR), which has a significant inhibitory effect on biological activity, has resulted in large amounts of BBR entering the environment.

5.On the other hand, if the author suggests degradation of berberine with microorganisms, is it safe, particularly if he describes using pathogenic strains e.g. Shigella sp. or Rhodococcus sp. (a few species are pathogenic).

In this paper, when berberine was degraded, most of the microorganisms with increased bacterial content were beneficial to the environment. Such as Methylotenera, Novosphingobium, Delftia and Shingopyxis.

6.In the Materials and methods section, the author shows that berberine was analyzed by HPLC method. Was the chromatographic method validated or was it implemented from another work? Please specify the appropriate reference.

This is a commonly used method in our lab. I have cited the article in our revised manuscript.

7.Please, place chromatograms for that analysis in the results section.

  We have made additions according to the reviewer’s suggestion.

8.Section 2.5. (here, the author wrote incorrectly 2.516 instead 2.5) from what samples the genome was extracted; it is not clear.

Corrected!  2.5. 16s rDNA sequencing of soil water samples. The genome of each sample (Xinjiang, Beijing, Jiangxi, Yunnan, Heilongjiang and Guangxi)

9.Section 2.6. - How bacteria were identified and characterized. Did the author use mass spectrometry? Please show the results for that analysis in the Results section.

We did the replicon sequencing. We supplied the results in the revised manuscript.

10.The author did not calculate any statistical analysis. Thus his inference is burdened with a significant error.

Data in the results were presented as means values ± standard errors of mean (SD). Statistical significances were indicated by P values analyzed by one-way ANOVA analysis.

11.The preparation of the manuscript is negligent. The methodology does not allow the repetition of the experiments.

We have made additions and modifications.

12.Section 3.5 is instead a review than a discussion. In this section, the author should present enzymatic or proteomic analysis results.

We have made additions and modifications.

Reviewer 2 Report

The work entitled: " Tetrahydroisoquinoline N-methyltransferase from Methylotenera is an essential enzyme for biodegradation of Berberine in soil water" presents an interesting aspect to evaluated the effect of BBR on the bacterial diversity of soil water, laying foundation for its application as a botanical pesticide. Hence, it deserves to be published after correcting some errors.

Here is the comments and suggestions:

1) Abstract should present a clear purpose and scope of the work, provide a short research methodology and present the most important conclusions. It needs to be corrected.

2) the subtitles of results section should be rephrased.

3) I couldn’t find the discussion section… since the 4th section entitled Conclusion!!

4) The conclusion should be more concise, generalising and summarizing.

5) English editing is highly recommended.

Author Response

1) Abstract should present a clear purpose and scope of the work, provide a short research methodology and present the most important conclusions. It needs to be corrected.

We agree with the comment and have made changes.

2) the subtitles of results section should be rephrased.

Corrected!

3) I couldn’t find the discussion section… since the 4th section entitled Conclusion!!

We agree with the comment and have made changes.

4) The conclusion should be more concise, generalising and summarizing.

We agree with the comment and have made changes.

5) English editing is highly recommended.

We agree with the comment and have made changes.

Round 2

Reviewer 1 Report

Generally, the author did not correct the manuscript. There were added single sentences, only.  

Section 2.5. 
The genome of each sample (Xinjiang, Beijing, Jiangxi, Yunnan, Heilongjiang and Guangxi) was extracted and used for replicon sequencing.

it is still not clear from what samples the genome was extracted

Section 2.6. 
After purification for three times, the genome of the strain was extracted and 16s rDNA was amplified and replicon sequenced.

The author added only replicon; 
If bacteria were identified on the basis of sequencing analysis, please provide relevant results in the results section and discuss these data.
In the results section, the author has shown only a list of bacteria in Table 1.

In Figure 3, the author has shown the content of bacteria;
How did the author calculate the content? It is not described in the Materials and method section.

Section 3.5 is rather a review than a discussion of the author's results.
The author describes: 
"The results indicated that orthologs of enzyme 1, 2 and 9 can be found in Methylotenera; Orthologs of enzyme 1, 3 and 24 were found in Novosphingobium; Orthologs of enzymes 1 and 3 were found in Delftia; Ortholog of enzyme 1 was found in Sphingopyxis and Flavobacterium (protein sequence similarity > 30%)".

However, this is in contradiction because the author did not present any results. 

Moreover, the first manuscript was poorly written and English changes were required. 
The author uploaded newly version without appropriate English corrections.

Author Response

Generally, the author did not correct the manuscript. There were added single sentences, only.  

We are sorry for this. We revised our manuscript carefully this time.

1) Section 2.5. 
The genome of each sample (Xinjiang, Beijing, Jiangxi, Yunnan, Heilongjiang and Guangxi) was extracted and used for replicon sequencing. it is still not clear from what samples the genome was extracted

The soil samples were taken from various places in China. After diluted in sterilized water and cultivated for BBR’s degradation, soil water was taken and the total bacterial genomes were extracted by using a genome extraction kit. Then the amplicon experiments were conducted and the extracted genomes were used as the template for the first round of PCR amplification of the V3V4 region of 16s rDNA.

2) Section 2.6. 
After purification for three times, the genome of the strain was extracted and 16s rDNA was amplified and replicon sequenced.

The author added only replicon; 
If bacteria were identified on the basis of sequencing analysis, please provide relevant results in the results section and discuss these data.
In the results section, the author has shown only a list of bacteria in Table 1.

We are sorry for the mistake. The experiments conducted in our work is ‘amplicon sequencing’ but not ‘replicon sequencing’ in our previous version of manuscript. We revised this throughout the manuscript.

3) In Figure 3, the author has shown the content of bacteria;
How did the author calculate the content? It is not described in the Materials and method section.

The content of each genus was calculated by comparing the amount of sequences identified belonging to certain genus and the total amount of sequences in the same sample. This is a standard method for quantification of bacterial content in amplicon experiment. We added details in our revised manuscript.

4)Section 3.5 is rather a review than a discussion of the author's results.
The author describes: 
"The results indicated that orthologs of enzyme 1, 2 and 9 can be found in Methylotenera; Orthologs of enzyme 1, 3 and 24 were found in Novosphingobium; Orthologs of enzymes 1 and 3 were found in Delftia; Ortholog of enzyme 1 was found in Sphingopyxis and Flavobacterium (protein sequence similarity > 30%)".

However, this is in contradiction because the author did not present any results. 

We revised our expression in this section. Indeed this statement is the result that we obtained by bioinformatics analysis.

5) Moreover, the first manuscript was poorly written and English changes were required. The author uploaded newly version without appropriate English corrections.

Thanks for the suggestion. We revised our manuscript according to the suggestions from a senior researcher.

Round 3

Reviewer 1 Report

Dear Editor,

In my opinion, the manuscript can be published in the Molecules journal. The authors eliminated most of the mistakes and corrected the English language.